**Data Availability Statement:** All relevant data are within the paper and its Supporting Information files.

# Effect of copaíba essential oil (*Copaifera officinalis* L.) as a natural preservative on the oxidation and shelf life of sheep burgers

Jéssica de Oliveira Monteschio[1]☯*, Fernando Miranda de Vargas Junior[2]‡, Adrielly Lais Alves da Silva[2]☯, Renata Alves das Chagas[2]‡, Tatiane Fernandes[2]‡, Ariadne Patricia Leonardo[2]‡, Isabelle Naemi Kaneko[3]☯, Laura Adriane de Moraes Pinto[4]‡, Ana Guerrero[5]☯, Antônio Alves de Melo Filho[6]☯, Vany Perpétua Ferraz[7]‡, Gisele Maria Fagundes[1]☯, James Pierre Muir[8]‡

1 Department of Animal Science, Federal University of Roraima, Boa Vista, Roraima, Brazil, 2 Department of Agrarian Sciences, Federal University of Grande Dourados, Dourados, Mato Grosso do Sul, Brazil, 3 Department of Animal Science, Federal University of Paraíba, Areia, Paraiba, Brazil, 4 Department of Animal Science, Federal University of Paraná, Palotina, Paraná, Brazil, 5 Departamento Producción y Sanidad Animal, Facultad de Veterinaria, Universidad Cardenal Herrera-CEU, CEU Universities, Alfara del Patriarca, Valencia, España, 6 Chemistry department, Federal University of Roraima, Boa Vista, Roraima, Brazil, 7 Chemistry department, Federal University of Minas Gerais, Belo Horizonte, Brazil, 8 Department of Soil and Crop Sciences, Tarleton State University, Stephenville, Texas, United States of America

☯ These authors contributed equally to this work.
‡ These authors also contributed equally to this work.
* jessica.monteschio@ufrr.br

## Abstract

We evaluated the effects of the inclusion of copaíba (*Copaifera officinalis* L.) essential oil at 0.05 and 0.1% as a possible replacement of synthetic additive butylated hydroxytoluene (BHT) in sheep burgers during 14 days of storage in a refrigerated display case (4°C). During the shelf life days, analyzes of antioxidant activity, lipid oxidation, pH, color, cooking loss, texture and consumer acceptability were carried out on refrigerated burgers. The addition of copaíba essential oil showed an antioxidant effect in sheep burgers ($P > 0.05$), reducing lipid oxidation. Copaiba essential oil added at 0.05% showed the highest antioxidant activity, decreased cooking losses and delaying discoloration (loss of redness) during storage; it further improved, the tenderness of sheep burgers ($P < 0.05$). The treatments had no effect ($P > 0.05$) on consumer acceptability. Copaiba essential oil is a promising natural antioxidant to increase the shelf life of meat products, as well as being a viable solution to replace synthetic antioxidant BHT, thereby promoting the fresh-like quality appeal of sheep burgers.

## Introduction

During storage, oxidative processes impact meat and meat products, mainly due to lipid oxidation, leading to deterioration. Oxidative processes adversely affect food taste, texture, nutritional value, and color, decreasing consumer acceptability [1, 2].

**Funding:** The author(s) received no specific funding for this work.

**Competing interests:** The authors have declared that no competing interests exist.

There is a need to produce foods with lower amounts of synthetic additives that, however, retain their original characteristics and long shelf life [3]. Butylated hydroxyanisole, butylated hydroxytoluene (BHT), propyl gallate, and *tert*-butylhydroquinone are the most common synthetic antioxidants used in the food industry [4]. However, there is increasing concern about the possible carcinogenic effects of synthetic antioxidants threatening consumer's health [5].

Consequently, there is increased focus on herbal extracts and essential oils as sources of natural antioxidants for food preservation [1]. Besides promoting the beneficial intake of healthier food containing functional ingredients, the incorporation of these natural antioxidants does not cause drastic changes in consumer eating habits. Natural antioxidants are promising substitutes for synthetic preservatives, and because of their presumed safety, they are preferred by health-conscious consumers [6].

Essential oils are natural plant products that exhibit antimicrobial and/or antioxidant properties [4, 7]. Many essential oils are considered "Generally Recognized as Safe" (GRAS) and approved for use in food by the United States Food and Drug Administration [8].

Copaíba (Copaifera spp.) essential oil (CO) is one of those essential oils approved as a flavoring agent in food and beverages [9] and, is widely distributed in northern South America, and produced in the Brazilian Amazon rainforest, and its main bioactive components are phenolic compounds, such as sesquiterpenes and diterpenes [10], the main component being β karyophylene, which constitutes more than 50% of the oil, selectively binds to the cannabinoid receptor 2 (CB2) and has been proposed as a viable strategy to relieve pain and inflammation [11, 12].

In commercial ovine herds focused on lamb production, ewes are generally used up to 6 or 7 years of age, and the older animals are discarded from production and slaughtered [13]. Adult sheep meat is underused and undervalued due to a lack of information. However, when the meat of these animals is processed, it can generate nutritionally valuable by-products that can be used to add value to products, improving consumer acceptance, and encouraging a better use of this meat by increasing the profitability of the sheep industry [6].

This study evaluated the potential of copaíba essential oil as a natural preservative, assessing its antioxidant potential to prevent/retard lipid oxidation in sheep meat burgers, evaluating its influence on the product quality (shelf life) and sensory characteristics (consumer acceptability) of sheep burgers during 14 days of refrigerated display.

## Material and methods

The sensory analysis study was approved by the Research Ethics Committee Involving Human Beings. Federal University of Roraima (protocol numbers: 23282819.9.0000.5302), Boa Vista, State of Roraima, Brazil.

### Characterization of Copaíba Essential Oil (CEO)

**Phytochemical screening by Gas Chromatography–Mass Spectrometry (GC-MS) and high-resolution gas chromatography (GC-FID).** Identification and quantification of the volatile constituents present in the CEO were performed by gas chromatography coupled to mass spectrometry (GC-MS), using a GC-MS (QP2010) ULTRA system (Shimadzu) with an Rxi-1MS (30 m x 0.25 mm x 0.25 µm) column (Restek) at temperatures ranging from 50°C (2 min), increasing at 3°C min$^{-1}$ to 230°C. The injection temperature was 250°C, and the injected volume was 1 µL in a split ratio of 1:10. The interface temperature between the chromatograph and the mass spectrometer (GC-MS) was 250°C and the electron ionization (EI) MS detector operated at an ionization energy of 70 eV at 250°C. Helium was used as carrier gas at a flow rate of 2.0 mL min$^{-1}$. The data acquisition GC-MS software Solution (Shimadzu) was used to

evaluate the components. The retention indices were compared with those of the National Institute of Standards and Technology (NIST) through the NIST11 mass spectral library based on the Kovats index, a linear retention index used to identify oil chemical constituents with calculations including the retention time of a series of n-alkanes. The HP 7820A (Agilent) gas chromatograph was used for the Gas Chromatography Flame Ionization Detector (GC-FID), with HP5 columns with dimensions of 30 m x 0.32 mm x 0.25 μm (Agilent), temperatures ranging from 50˚C (2 min), with increases of 3˚C min$^{-1}$ up to 230˚C. The injection temperature was 250˚C in a split proportion of 1:10, and the FID temperature was 250˚C. Helium was used as carrier gas at 3 mL min-1 and an injection volume of 1 μL. The data acquisition software used was EZChrom Elite Compact (Agilent). The analyses were performed in triplicate, and quantitative data were obtained by electronic integration of the peak area in relation to the total area of the chromatogram, which resulted in the concentration (%) of each chemical constituents present in the essential oil, according to Moniz et al. 2019 [14].

## Preparation of burgers

The burgers were divided into four treatment groups: control without antioxidants (CON), burgers containing synthetic additive (BHT), burgers with 0.05% CEO (CEO-0.05%), and burgers with 0.1% CEO (CEO-0.1%).

The sheep burgers (40 g) were distributed to the different treatments (two replicates of experiment with three samples by each analysis day). Burgers were made from shoulder and hind limb cuts of discarded Suffolk sheep ($> 7$ years) with an average body weight of 55 kg and a body condition score of 3.5 points (on a scale of 1–5 scale). These cuts were deboned, and the edible portions (meat and fat) were separated and ground, added to the mix 10 g NaCl/kg meat for all treatments and 50 mg/kg of BHT for BHT treatments according to Fernandes et al. 2016 [6]. These authors, obtained satisfactory results with this concentration, not exceeding the limit of 100 mg/kg for BHT, which is the maximum concentration allowed in meat products, established by the Ministry of Health of Brazil.

We added 0.05% copaíba essential oil/kg of meat for the CEO-treatment 0.05% and 0.1%/kg for the CEO-treatment 0.1%. The essential oils were added directly to the meat and salt mixture, followed by weighing and preparation of the burgers in molds. The CEO was purchased from Ferquima® (Vargem Grande Paulista, São Paulo, Brazil) and had been extracted by steam distillation of the copaíba oleoresin, extracted directly from the stem of *Copaifera officinalis* L. The essential oil was tested for purity, density, and toxicity and incorporated into the burgers. The concentrations used were based on previous studies [4, 15] that obtained satisfactory results in terms of antioxidant activity. The main compounds in CEO are described in (Fig 1).

The sheep burgers were produced in molds of 10-cm diameter and 1-cm height, specific for the sheep burger preparation, and packed in polystyrene trays covered with polyethylene film permeable to oxygen. The trays were placed in a refrigerated display case (4˚C), under fluorescent light (380 lux, 12 h/day), simulating typical conditions of the Brazilian market. Analyses were performed at 1, 3, 7, and 14 days of storage.

## Antioxidant activity

**Burger bioactive compound extracts.** Burger extracts (1:1 w/v with methanol) were obtained according to de Oliveira Monteschio. 2017 [7], using an Ultra-Turrax homogenizer, followed by centrifugation and filtration. The extracts were analyzed for antioxidant activity by measuring the 2,2-diphenyl-1-picrylhydrazyl (DPPH) and 2,2'-azino-bis(3-ethylbenzothiazoline-6-sulfonic) (ABTS) radical scavenging activities, as well as the total phenolic content (TPC).

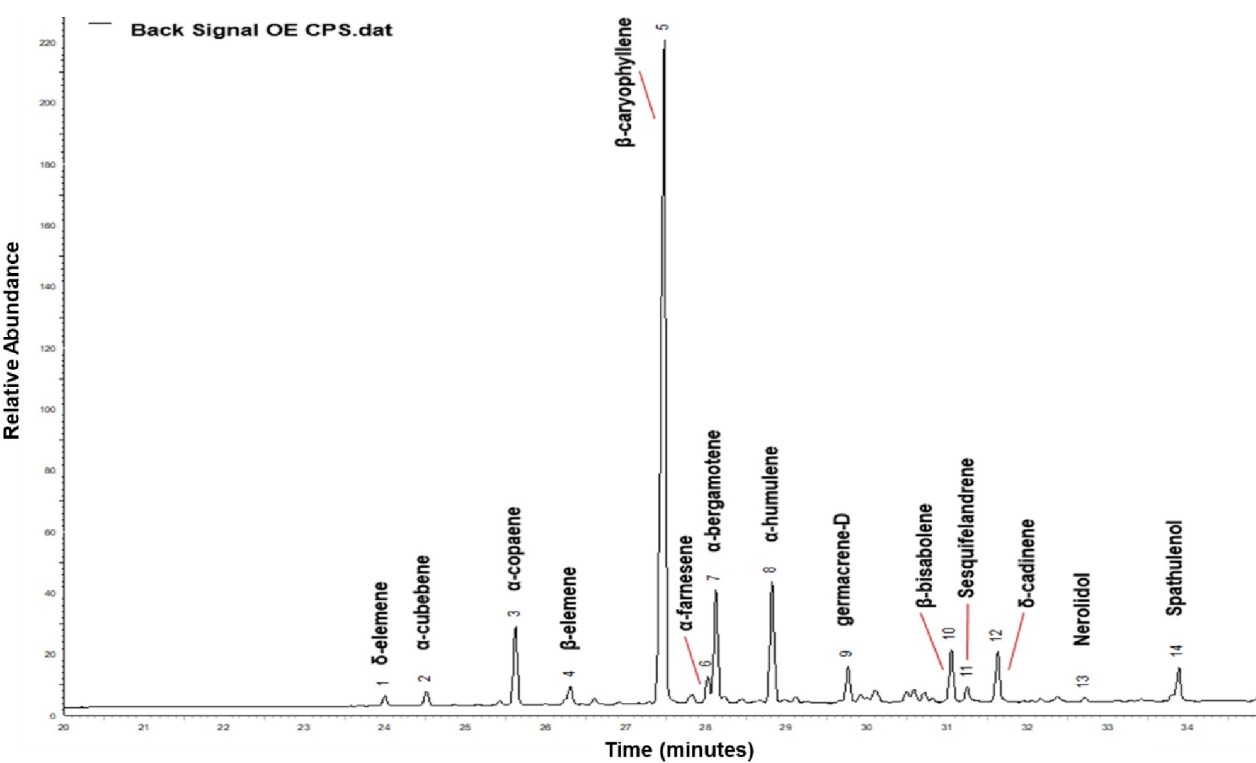

**Fig 1. Concentration of chemical components in essential oil of Copaiba.**

DPPH radical scavenging assay. The DPPH radical scavenging activity was measured according to Li et al. 2009 [16], with modifications. Meat extract (150 μL) was mixed with 2850 μL of a methanolic DPPH (60 μM) solution and reacted for 30 min. Absorbance at 515 nm was measured against pure methanol. Antioxidant activity was calculated as DPPH radical scavenging activity (%) = (1- ($A_{sample\ t0}$/$A_{sample\ t}$) × 100, where $A_{sample\ t0}$ is the absorbance of the sample at time zero, and $A_{sample\ t}$ is the absorbance of the sample at 30 min.

**ABTS radical scavenging assay.** The ABTS assay was conducted according to Re et al. 1999 [17] with modifications. ABTS+• was generated by the reaction of 7 mM ABTS (5 mL) with 140 mM potassium persulfate (88 μL), and the mixture was incubated in the dark at 25˚C for 16 h. The ABTS-activated radical was diluted with ethanol to an absorbance of 0.70 ± 0.02 at 734 nm. The radical scavenging activity (%) was also measured at 734 nm. Meat extract (40 μL) was mixed with ABTS+• solution (1,960 μL), and the absorbance was recorded after 6 min. The ABTS radical scavenging activity (%) was calculated as 1-($A_{sample\ t0}$/$A_{sample\ t}$) × 100, where $A_{sample\ t0}$ is the absorbance of the sample at time zero and $A_{sample\ t}$ is the absorbance of the sample at 6 min.

**Folin–Ciocalteu assay for Total Phenolic Content (TPC).** The TPC was determined according to Singleton & Rossi 1965 [18], with modifications. An aliquot of each extract (125 mL) was mixed with 125 mL of Folin–Ciocalteu reagent (1:1 v/v deionized water) and 2,250 mL $Na_2CO_3$ (28 g/L). The solutions were then incubated in the dark at 25˚C for 30 min, and absorbance was measured at 725 nm using a spectrophotometer (Evolution 201 UV–visible spectrophotometer, Thermo Scientific). Results were expressed as milligrams of gallic acid equivalents (GAE) per gram of sample. A gallic acid standard curve (0–300 mg/L) was prepared.

## Lipid oxidation (thiobarbituric acid-reactive substances, TBARS)

The malondialdehyde (MDA) content in burger was measured by the thiobarbituric acid-reactive substances assay [19]. The sample (5 g) was mixed with 15 mL of TCA solution (7.5% trichloroacetic acid, 0.1% gallic acid, and 0.1% EDTA), homogenized using an Ultra-Turrax, then centrifuged for 15 min. The supernatant was filtered and mixed with TBA solution (1% thiobarbituric acid, 15% trichloroacetic acid, and 562.5 µM HCl). The obtained mixture was boiled (100˚C) for 15 min, cooled, and absorbance at 535 nm was compared with an MDA standard. Results were expressed as mg MDA/kg of meat.

## pH measurements

The pH was measured at 1, 3, 7, and 14 days of display, using a pH-meter with a Testo 205/206 penetration probe. Prior to use, the pH meter was calibrated at 20˚C using standard pH 4.0 and 7.0 buffers.

## Instrumental meat color

The CIELab color parameters were recorded using a Minolta CR-400 chroma meter (Japan) under D65 illumination, aperture of 8 mm, and a closed cone set on the $L^*a^*b^*$ system, with a 10˚ view angle. Six measurements at randomly selected points were recorded per sample, obtaining lightness ($L^*$), redness ($a^*$), and yellowness ($b^*$).

## Cooking losses

The burgers at 1, 3, 7, and 14 days of display were weighed and wrapped in aluminum foil. Each sample was cooked in a pre-heated grill (Grill Philco Jumbo Inox, Philco SA, Brazil) at 200˚C until the internal temperature reached 72˚C, which was monitored using an Incoterm internal thermocouple (145 mm; Incoterm LTDA, Brazil). Once internal burger temperature reached 25˚C, each burger was weighed. The cooking weight losses were calculated for each day as a percentage relative to the initial weight, according to the following equation:

$$CL\ (\%) = \frac{Wi - Wf}{Wi \times 100} \tag{1}$$

where $CL$ is the cooking loss $Wi$ is the initial weight (at the day of production), and $Wf$ is the final weight (at 1, 3, 7, and 14 days of storage, respectively).

## Texture measurement

Burgers were removed from the heat and left at ambient temperature to cool to 25˚C. Subsequently, they were cut into rectangular pieces of 1-cm² cross-sections (eight pieces per treatment). Texture was analyzed using a texture analyzer installed with a Warner–Bratzler blade and set with a 50-kg load cell and an operating speed of 2 mm/s. The maximum shear force (kg) was recorded at 1, 3, 7, and 14 days of display.

## Consumer acceptability

Participants filled out a written consent form through which they were informed about all questions regarding the procedures that would be performed in the research and through this information they could decide whether they would be volunteers for the study. The term also exempt the participant of any liability with respect to your health.

Seven sessions were carried out in a room adequate to perform sensory analyses. Each session had 10 different consumers, and each consumer evaluated four burger samples, one per

treatment. A randomized design was used to serve the meat, avoiding carry-over and order effects. The burgers were individually packed with aluminum foil and cooked. Seventy consumers participated in the sensory test; however, three consumers with missing data and outliers in the questionnaire were not considered for statistical analyses. The consumers were untrained tasters randomly selected from the university attendees (students, employees, and visitors) whose sociodemographic profile fit the Brazilian national profile quotas of gender (36 men and 31 women) and age (from 18 to 70 years). The burgers were cooked on a pre-heated grill at 200˚C until the internal temperature reached 70˚C. Each burger was cut into four triangles and kept warm (50˚C) until consumer evaluation (around 10 min after cooking).

The acceptability of the burger at Day 1 of storage was evaluated for the following attributes: odor, tenderness, flavor, and overall acceptability. Ratings were scored using a structured hedonic nine-point scale (1 = dislike extremely; 9 = like extremely), without the middle level.

## Statistical analyses

Statistical analyses were performed using SPSS (Version 23.0; IBM SPSS Statistics, SPSS Inc., Chicago, IL, USA) for Windows. A fully randomized factorial design was implemented, with four treatments (CON, BHT, CEO-0.05%, and CEO-0.1%) and four storage times (1, 3, 7, and 14 days). Treatment and storage time were considered fixed factors in a factorial design. For consumer acceptability, treatment was the only fixed effect evaluated and consumer was considered a random effect and session a blocking effect. Differences were considered significant at $P \leq 0.05$. Tukey's test was performed.

## Results

### Phytochemical screening by GC-MS and GC-FID

Fourteen compounds were identified in the CEO samples, representing 90.9% of the total oil. The main constituents of CEO were β-caryophyllene (51.1%), α-humulene (8.1%), and α-bergamotene (7.4%). Sesquiterpene hydrocarbons constituted the largest fraction of the oil (88.1%), and oxygenated sesquiterpenes reached 2.8% (Table 1).

### Total Phenolic Content (TPC) and antioxidant activity

Three methods (ABTS, DPPH, and TPC) were used to verify if the antioxidant activity of the burgers was improved by the incorporation of CEO. All three trials showed similar results (Fig 2).

The TPC antioxidant activity for sheep burgers decreased significantly ($P < 0.05$) during cold storage for CON and BHT treatments. Higher values were observed by DPPH assay at Day 3 of storage ($P < 0.05$) for CEO-0.05%, intermediate values for BHT, and lower values for CON and CEO-0.1%.

In the ABTS assay, similar activities were observed, showing a decrease ($P < 0.05$) during cold storage for CON, BHT, and CEO-0.1% treatments. Differences among treatments ($P < 0.05$) were observed at Day 7 of storage, with greater antioxidant activity found in CEO-0.05% burgers relative to the other treatments.

### Lipid oxidation

Lipid oxidation of sheep burgers, as measured by MDA production, was affected by treatments ($P < 0.05$) and increased with aging (1, 3, 7, and 14 days) (Fig 3). On Day 7 of storage, treatments with CEO (CEO-0.05% and CEO-0.1%) presented lower MDA values among the studied treatments, and the greatest MDA contents were detected in CON. At Day 14 of storage, CEO-0.05% presented the lowest MDA content, while CON had the highest.

**Table 1. Identification and concentration of chemical components in essential oil of Copaiba.**

| Probable substance | RI* | Concentration (%) |
|---|---|---|
| **Sesquiterpene hydrocarbons** | | |
| δ-elemene | 1340 | 0.7 |
| α-cubebene | 1353 | 1.0 |
| α-copaene | 1381 | 5.1 |
| β-elemene | 1399 | 1.6 |
| β-caryophyllene | 1428 | 51.1 |
| α-farnesene | 1442 | 1.8 |
| α-bergamotene | 1444 | 7.4 |
| α-humulene | 1462 | 8.1 |
| germacrene-D | 1486 | 2.5 |
| β-bisabolene | 1519 | 3.7 |
| Sesquiphellandrene | 1524 | 1.4 |
| δ-cadinene | 1533 | 3.7 |
| **Oxygenated sesquiterpenes** | | |
| Nerolidol | 1561 | 0.4 |
| Spathulenol | 1590 | 2.4 |
| **Others** | | 9.1 |
| **Total** | | 100.0 |

*Retention index.

## Color parameters

The color values ($L^*$, $a^*$ and $b^*$) associated with treatments and storage time are given in (Table 2). Lightness ($L^*$) was different among treatments ($P < 0.05$) on Day 7, with CEO-0.1% presenting the greatest value, followed by CEO-0.05%, while CON and BHT sheep burgers showed the lowest values. Storage time decreased the $L^*$ values ($P < 0.05$) of the BHT and CEO-0.1% samples.

There was a notable difference in the $a^*$ values among sheep burgers ($P < 0.05$) at Day 7 of storage, with redness intensity being greatest for CEO-0.05%, followed by CEO-0.1%; CON and BHT exhibited the lowest redness intensity. All treatments showed a decrease in $a^*$ (redness) values as storage progressed, but the lowest $a^*$ ($P < 0.05$) value at the final evaluation corresponded to burgers treated with the natural antioxidant CEO.

At the beginning of storage, CEO-0.1% displayed greater $b^*$ values (yellowness; $P < 0.05$) than CON and CEO-0.05%, while BHT burgers presented the least yellowness. However, during storage, yellow intensity ($b^*$) decreased, with declines ($P > 0.05$) observed for CEO-0.05% and CEO-0.1% at Day 14.

## pH, cooking loss, and texture

The addition of natural antioxidant or BHT did not induce differences ($P > 0.05$) in pH among treatments. However, pH changes ($P < 0.05$) occurred during storage, with an increase in pH values for CON, BHT, and CEO-0.05% treatments, and higher values observed at Day 14 compared with shorter storage durations (Table 3).

At Days 7 and 14 of storage, treatments affected ($P < 0.05$) cooking losses (Table 3). On Day 7, cooking loss was greatest in burgers treated withCEO-0.1% least for CEO-0.05%, while CON and BHT had intermediate losses. On Day 14, CEO-0.05%-treated burgers presented the least cooking loss, followed by BHT, CEO-0.1%, and CON recorded the largest cooking loss.

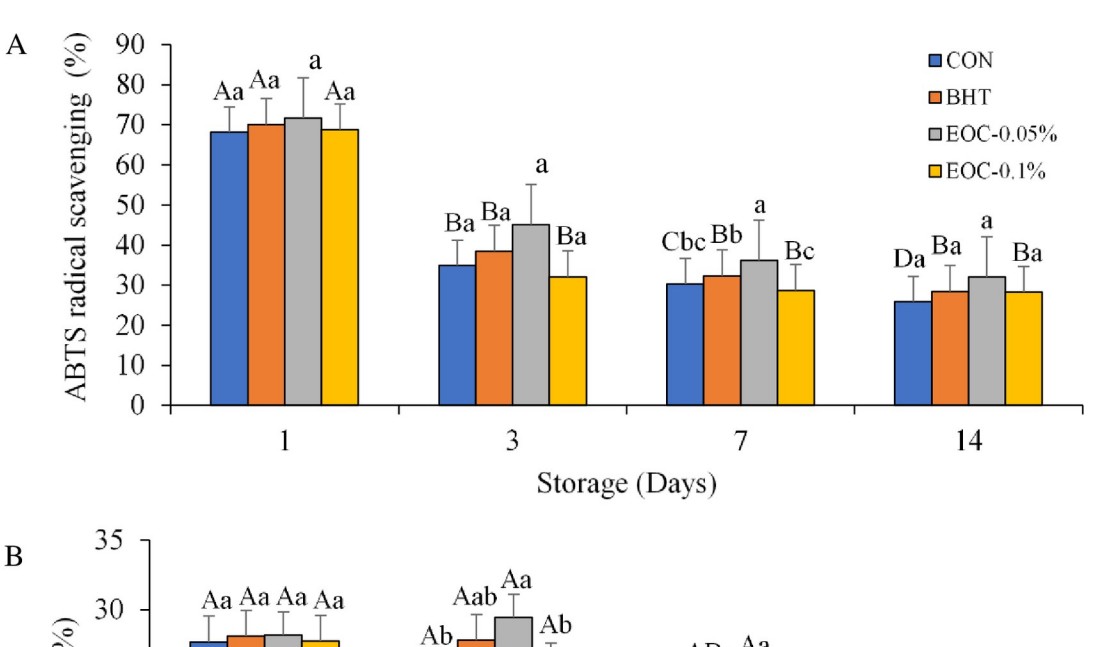

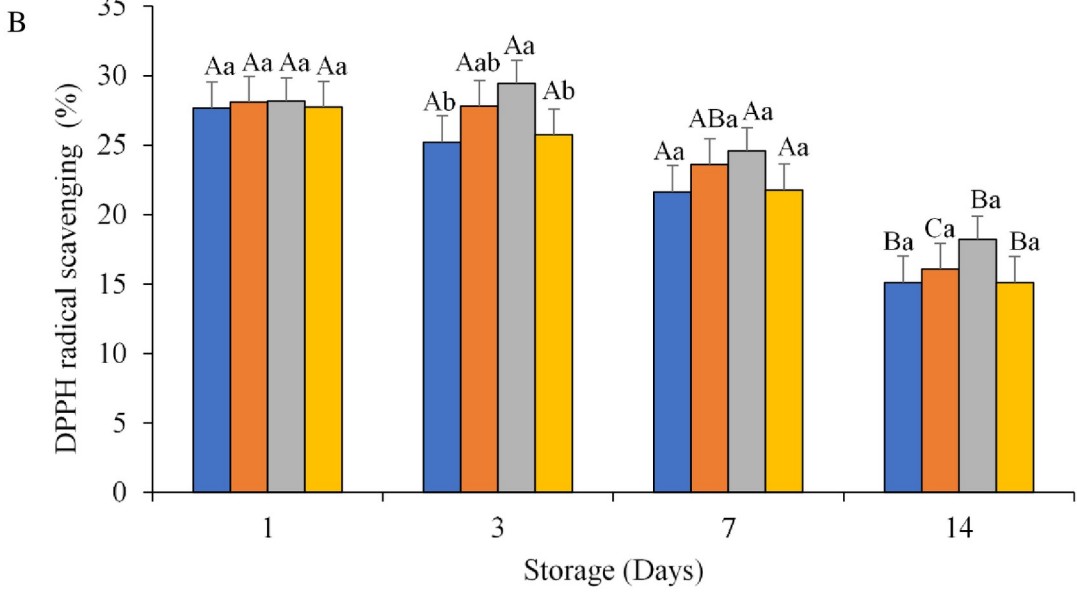

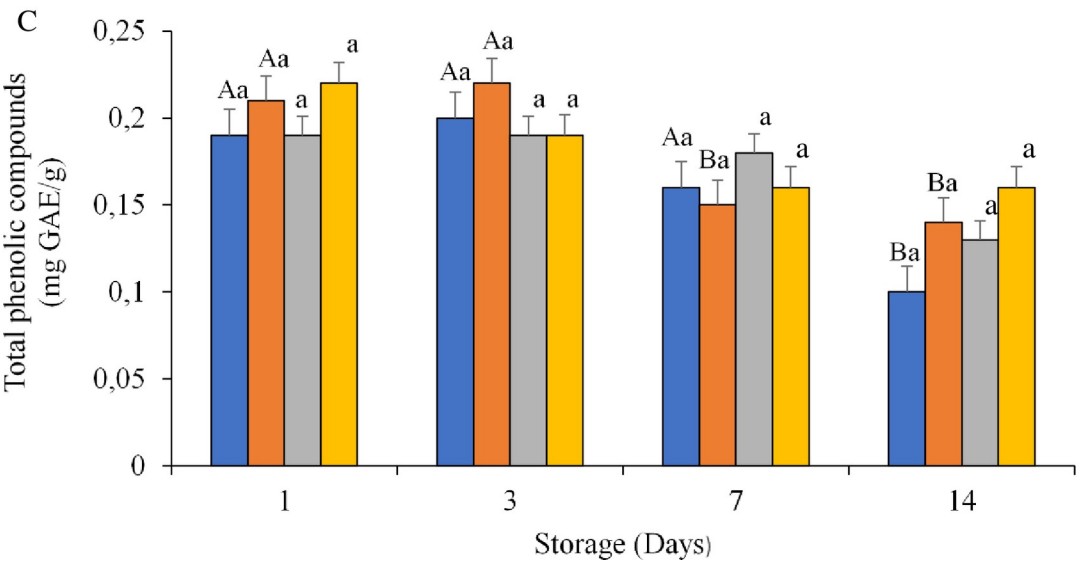

**Fig 2. Antioxidant activity ABTS, DPPH radical scavenging and Total phenolic compounds of sheep burgers during storage.** CON–sheep burger, without antioxidants; BHT—burger containing hydroxytolueneobutylated; CEO-0.05% sheep burger with 0.05% of Copaiba essential oil; CEO-0.1%: sheep burger with 0.1% of Copaiba essential oil). Different lower case letters in the same line are significantly different between treatments. Different upper case letters in the same column are significantly different between days of storage in a treatment.

Differences ($P < 0.05$) were found among storage periods, with a decrease in cooking loss as storage duration increased.

The treatments also affected ($P < 0.05$) the tenderness of the burgers on Day 14 of storage (Table 3). CEO-0.05% was the most tender burger, while BHT was the least tender one; CON and CEO-0.1% had intermediate values.

## Consumer acceptability

The sensory evaluation of sheep burgers was not affected by the formulation tested (Table 4). Consumers were unable to distinguish ($P > 0.05$) differences in odor, taste, texture, and overall acceptability among burgers despite the differences in antioxidant type and concentration (Table 4).

## Discussion

### Phytochemical screening by GC-MS and GC-FID

Plant essential oils differ in their antioxidant and antibacterial compounds [2]. Based on the results obtained in this study, CEO showed antioxidant properties. Its major constituent was

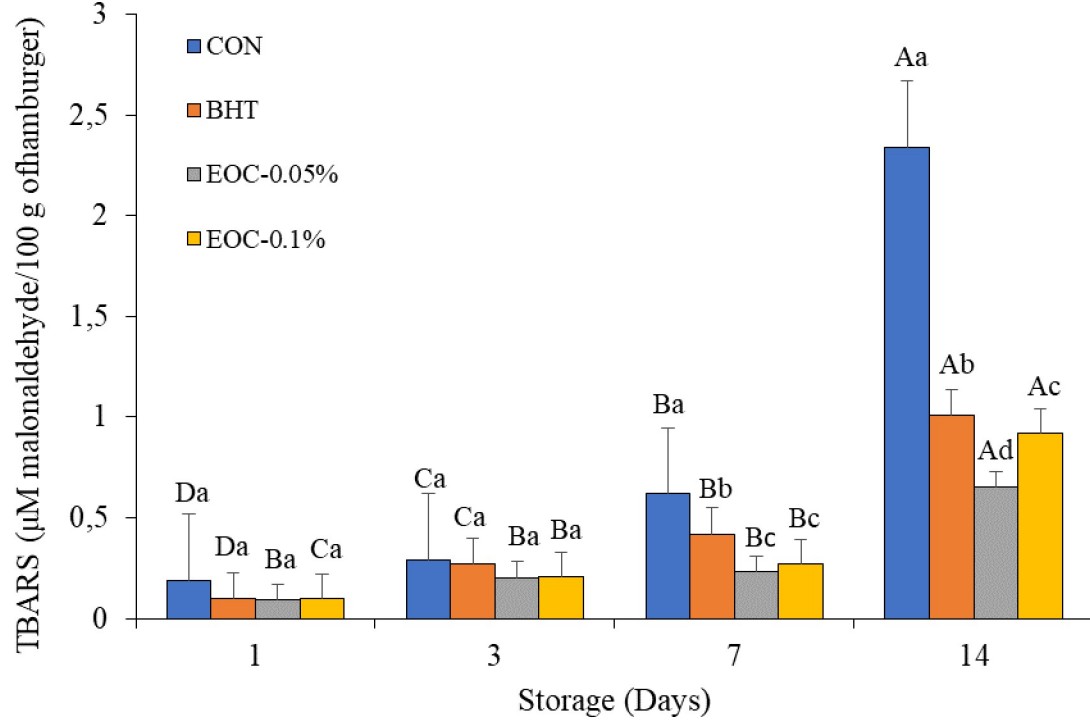

**Fig 3. Effect of copaiba essential oil and BHT inclusion on lipid oxidation of sheep burger during storage.** CON–sheep burger, without antioxidants; BHT—burger containing hydroxytolueneobutylated; CEO-0.05% sheep burger with 0.05% of Copaiba essential oil; CEO-0.1%: sheep burger with 0.1% of Copaiba essential oil). Different lower case letters in the same line are significantly different between treatments. Different upper case letters in the same column are significantly different between days of storage in a treatment.

**Table 2. Effect of Copaiba essential oil and BHT inclusion on color changes of sheep burgers during storage.**

| | Diets | | | | SEM[5] | P< |
|---|---|---|---|---|---|---|
| | CON[1] | BHT[2] | EOC-0.05%[3] | EOC-0.1%[4] | | |
| L* | | | | | | |
| 1 | 34.53 | 35.40A | 34.88 | 37.12A | 0.832 | 0.797 |
| 3 | 32.17 | 33.40AB | 34.70 | 36.26AB | 0.730 | 0.235 |
| 7 | 32.03b | 32.20ABb | 33.56ab | 35.46ABa | 0.542 | 0.012 |
| 14 | 30.75 | 30.97B | 30.06 | 30.89B | 0.362 | 0.878 |
| SEM | 0.852 | 0.655 | 0.875 | 0.990 | | |
| P< | 0.569 | 0.023 | 0.150 | 0.043 | | |
| a* | | | | | | |
| 1 | 12.34A | 12.29A | 11.89A | 12.51A | 0.266 | 0.450 |
| 3 | 10.55A | 11.13A | 11.44A | 11.16AB | 0.175 | 0.402 |
| 7 | 9.19Ab | 9.64Bb | 11.87Aa | 10.57Bab | 0.413 | 0.026 |
| 14 | 4.75B | 5.19C | 6.98B | 5.25C | 0.375 | 0.118 |
| SEM | 1.08 | 0.934 | 0.811 | 1.047 | | |
| P< | 0.003 | 0.000 | 0.010 | 0.000 | | |
| b* | | | | | | |
| 1 | 4.10ab | 3.80b | 4.69Aab | 5.38ABa | 0.249 | 0.050 |
| 3 | 4.17 | 4.99 | 5.68A | 6.10A | 0.326 | 0.132 |
| 7 | 3.60 | 3.72 | 5.17A | 4.50AB | 0.278 | 0.113 |
| 14 | 2.66 | 3.32 | 3.18B | 3.08B | 0.169 | 0.661 |
| SEM | 0.278 | 0.263 | 0.360 | 0.464 | | |
| P< | 0.183 | 0.069 | 0.003 | 0.047 | | |

Different lower case letters in the same line are significantly different. Different upper case letters in the same column are significantly different.

[1]CON–sheep burger, without antioxidants

[2]BHT—burger containing hydroxytolueneobutylated

[3]CEO-0.05% sheep burger with 0.05% of Copaiba essential oil

[4]CEO-0.01%: sheep burger with 0.01% of Copaiba essential oil.

[5]SEM: Standard error of means.

β-caryophyllene (51.1%), which is a terpenoid-type sesquiterpenes common in essential oil of several plants [20]. This compound has proven effects as an anti-inflammatory, local anesthetic, anti-carcinogenic, antioxidant, antimicrobial, and neuroprotective agent [21].

## Total Phenolic Content (TPC) and antioxidant activity of sheep burgers

The antioxidant properties of the plant are mainly due to the contents of phenolic compounds that can donate a hydrogen atom to free radicals and thereby delay or inhibit lipid oxidation, which is a major problem in food processing [22]. Antioxidants may also prevent free radical-induced cell damage [23].

As mentioned earlier, although BHT is widely used as an antioxidant in food, its consumption has been linked to cancer, asthma, and behavioral disorders in children [24]. The present study confirms that it is possible to confer antioxidant activity to food by the addition of CEO as a natural compound. The same observation was made by Kempinski et al. 2017 [4] when evaluating oregano essential oil.

When de Carvalho et al. 2019 [25] investigated the effects of guarana (*Paullinia cupana*) and pitanga (*Eugenia uniflora* L.) leaf extracts on lamb burger, antioxidant effects decreased over time (from 0 to 18 days). A similar behavior was found in this study.

**Table 3. Effect of Copaiba essential oil and BHT inclusion on quality parameters pH, cooking losses and shear force in sheep burgers during storage.**

| Variable | CON[1] | BHT[2] | EOC-0.05%[3] | EOC-0.1%[4] | SEM[5] | P< |
|---|---|---|---|---|---|---|
| **pH** | | | | | | |
| 1 | 6.02B | 6.03B | 6.00C | 6.02 | 0.015 | 0.960 |
| 3 | 6.05B | 6.05B | 6.02C | 6.03 | 0.005 | 0.158 |
| 7 | 6.04B | 6.06B | 6.08B | 6.08 | 0.007 | 0.283 |
| 14 | 6.47A | 6.36A | 6.30A | 6.35 | 0.036 | 0.490 |
| SEM | 0.072 | 0.052 | 0.045 | 0.057 | | |
| P< | 0.003 | 0.005 | <0.001 | 0.070 | | |
| **Cooking loss, %** | | | | | | |
| 1 | 37.54A | 38.32A | 39.39A | 38.47A | 0.276 | 0.061 |
| 3 | 33.40C | 32.29B | 32.80B | 33.72C | 0.228 | 0.051 |
| 7 | 34.93Bab | 33.64Bab | 33.16Bb | 35.33Ba | 0.364 | 0.036 |
| 14 | 28.88Da | 25.52Cb | 23.23Cc | 24.48Dbc | 0.794 | <0.001 |
| SEM | 1.190 | 1.735 | 2.188 | 1.970 | | |
| P< | <0.001 | <0.001 | <0.001 | <0.001 | | |
| **Shear force, kg** | | | | | | |
| 1 | 1.39 | 1.42 | 1.39 | 1.41 | 0.012 | 0.883 |
| 3 | 1.27 | 1.32 | 1.21 | 1.39 | 0.081 | 0.930 |
| 7 | 1.24 | 1.31 | 1.10 | 1.20 | 0.063 | 0.802 |
| 14 | 1.20ab | 1.30a | 0.84b | 1.09ab | 0.073 | 0.043 |
| SEM | 0.402 | 0.020 | 0.107 | 0.078 | | |
| P< | 0.429 | 0.069 | 0.399 | 0.520 | | |

Different lower case letters in the same line are significantly different. Different upper case letters in the same column are significantly different

[1]CON–sheep burger, without antioxidants

[2]BHT—burger containing hydroxytolueneobutylated

[3]CEO-0.05% sheep burger with 0.05% of Copaiba essential oil

[4]CEO-0.01%: sheep burger with 0.01% of Copaiba essential oil.

[5]SEM: Standard error of means.

**Table 4. Effect of Copaiba essential oil and BHT inclusion on consumer acceptability of sheep burgers during storage (n = 67 consumers)[§].**

| Variable | CON[1] | BHT[2] | EOC-0.05%[3] | EOC-0.1%[4] | SEM[5] | P< |
|---|---|---|---|---|---|---|
| **Odor** | 6.70 | 6.88 | 7.01 | 7.02 | 0.095 | 0.590 |
| **Flavor** | 6.89 | 7.22 | 7.26 | 7.28 | 0.092 | 0.401 |
| **Tenderness** | 7.29 | 7.26 | 7.28 | 7.40 | 0.092 | 0.955 |
| **Overall** | 6.95 | 7.17 | 7.28 | 7.29 | 0.089 | 0.510 |

[§] Based on a 9-point scale (1: dislike extremely; 9: like extremely). Different lower case letters in the same line are significantly different.–sheep burger, without antioxidants

[2]BHT—burger containing hydroxytolueneobutylated

[3]CEO-0.05% sheep burger with 0.05% of Copaiba essential oil

[4]CEO-0.01%: sheep burger with 0.01% of Copaiba essential oil.

[5]SEM: Standard error of means.

## Changes in lipid oxidation during sheep burger storage

The increase in lipid oxidation was more pronounced for CON, probably due to the relative lack of phenolic compounds and the antioxidant activity of the CEO and BHT. The antioxidant activity of essential oils is a result of their phenolic and flavonoid compounds, which can donate a hydrogen atom in oxidation reactions, eliminate free radicals, and complex metal ions, mainly copper and iron, which catalyze lipid oxidation [26]. Food oxidation tends to increase with shelf life, and synthetic antioxidants (such as BHT) are used to slow down this process, as oxidation is one of the major factors responsible for food spoilage and leads to consumer rejection [27].

In this study, the use of the natural antioxidant CEO at 0.05 and 0.1% was more effective than that of BHT at delaying lipid oxidation in burgers during storage for 14 days. These results highlight the effectiveness of antioxidants, especially natural antioxidants, and corroborate the work of other authors [25, 28]. Such studies argued that natural extracts have greater antioxidant activity than synthetic antioxidants, suggesting the possibility of using these extracts as substitutes for commercial additives. The stability of lipid oxidation depends on the balance between antioxidant and prooxidant components [29].

In similar studies [2, 22], lipid oxidation of burger during refrigerated storage for up to 20 days declined with the incorporation of essential oils. This observation is consistent with the effect of CEO, evidenced in this work. Other studies have also demonstrated the effectiveness of natural compounds in retarding lipid oxidation during storage [1, 30].

## Changes in color parameters during sheep burgers storage

According to Olivo et al. 2001 [31], the color of the meat surface is the result of selective absorption of light by myoglobin and other important components, such as muscle fibers and their proteins, and is also influenced by the amount of free liquid present in the meat.

Teixeira et al.2013 [32] observed a dramatic decrease in the luminosity ($L^*$) of burgers formulated with moringa (*Moringa oleifera*) leaf flour as a natural antioxidant when applied to the meat surface. Likewise, the CEO-0.1% treatment had the same impact in this experiment. In the elaboration of sheep meat burger with oregano (*Origanum vulgare*) extract, Fernandes et al. 2016 [6] found no effects following 7 days of storage but, as with the trend seen in the present study, $L^*$ decreased for the burgers incorporated with synthetic and natural antioxidant, respectively, up to Day 20 of storage.

Most studies that instrumentally measured flesh color focused on the value of $a^*$ because redness is an important component in the visual appeal of meat by consumers [33]. The greater red intensity found for burgers containing CEO can be explained by the lower oxidation of the heme group within the iron atom and lower metmyoglobin formation [34] because of the action of antioxidants added (CEO-0.05%, and CEO-0.1%). Antioxidants contribute to color stabilization by delaying discoloration [35], as confirmed in other studies [1, 6, 25].

The high $b^*$ value of the burger treated with CEO-0.05% and CEO-0.1% at the beginning of the experiment can be attributed to the color of CEO, which varies from light yellow to medium yellow, golden yellow, and brown [36]. The same increase in $b^*$ value was recorded when adding curcumin (derived from *Curcuma longa)* to an oil-enriched meat product formulation [4]. Likewise, the addition of oregano as a natural antioxidant decreased the yellowing ($b^*$) of meat samples during storage [1]. Fernandes et al. 2017 [28] also described a reduction in yellow intensity during storage of hamburgers treated with natural antioxidants.

## Determination of pH value, cooking loss and texture

The slight increase in pH during storage might have been caused by the accumulation of volatile bases (e.g., ammonia and trimethylamine) generated during microbial or enzymatic amino

acid degradation [35]. Similar results have been found by Ghaderi-Ghahfarokhi et al. 2017 [37]. For instance, pH values increased throughout storage in beef patties to which cinnamon (*Cinnamomum verum*) had been added [38].

During storage, water loss is expected as a consequence of changes in muscle fibers caused by rigor mortis and changes in myofibrillar structure [39], which was observed in the trial. Burgers treated with CEO-0.1% showed high cooking loss values at Day 7 of storage, which can be explained by the active principle of each essential oil and its dosage. Essential oils can become toxic to cells and cell membranes due to prooxidant effects, causing an increase in water loss. The same phenomenon was observed by Rivaroli et al. 2016 [39] when evaluating an essential oil blend (oregano, garlic (*Allum ativum*), lemon (*Citrus limon*), rosemary (*Salvia rosmarinus*), thyme (*Thymus vulgaris*), eucalyptus (*Eucalyptus globulus*), and sweet orange (*Citrus sinensis*)) in meat.

Incorporating antioxidants into meat by the addition of essential oils may decrease the oxidation of cysteine proteases, such as calpain [40]. When oxidized, calpain loses most of its activity, and this suppresses oxidative processes by improving proteolysis and leading to meat softening [41]. In this trial, the lower burger oxidation conferred by the CEO-0.05% treatment might explain the greater tenderness of the meat relative to the other treatments.

### Consumer acceptability

The incorporation of CEO in the preparation of sheep burgers did not affect the sensory characteristics of the product, asserting the benefits of plant extracts as natural antioxidants [1, 25, 30]. Similarly, Sharafati-Chaleshtori et al. 2014 [2] and Sharafati Chaleshtori et al. 2015 [22] improved the overall acceptability of beef burgers by incorporating the essential oils of tarragon (*Artemisia dracunculus*) and basil (*Ocimum basilicum*). Fernandes et al. 2016 [6] did not initially observe differences among treatments for redness, surface discoloration, and unpleasant odor when evaluating the inclusion of oregano extract in the elaboration of sheep meat burger, the same species evaluated in this trial.

The addition of essential oils prevents the formation of secondary oxidation compounds and, consequently, decreases the rate of taste deterioration, increasing acceptability indices [42].

### Conclusion

The 0.05% concentration of copaíba essential oil was the most efficient one among the evaluated concentrations and is a promising natural alternative for future applications. The best antioxidant activity, reduction of lipid oxidation, and delayed discoloration, together with the sensory properties observed, suggest that the formulation of meat foods with the addition of copaíba essential oil can be explored as a new strategy for the development of more natural and healthier products than foods with synthetic antioxidants. We therefore conclude that *Copaifera officinalis* L. is a promising candidate as an antioxidant agent and to increase the shelf life of meat products with a high lipid content, which can be studied and applied for purposes in the food industry. However, new studies must be carried out so that this oil can be widely applied in the food industry, considering that there are no other studies in which the essential oil of copaíba has been used for the purpose suggested by this research.

### Supporting information

**S1 File. Supplementary data.** (color data, pH data, cooking loss data,texture data, consumer acceptability data, lipid oxidation data, total phenolic content data and antioxidant activity

data).
(XLSX)

**S1 Data.**
(XLSX)

## Author Contributions

**Conceptualization:** Jéssica de Oliveira Monteschio.

**Investigation:** Fernando Miranda de Vargas Junior, Vany Perpétua Ferraz.

**Methodology:** Adrielly Lais Alves da Silva, Renata Alves das Chagas, Tatiane Fernandes, Ariadne Patricia Leonardo, Antônio Alves de Melo Filho, Vany Perpétua Ferraz.

**Project administration:** Jéssica de Oliveira Monteschio, Fernando Miranda de Vargas Junior.

**Supervision:** Fernando Miranda de Vargas Junior, Antônio Alves de Melo Filho.

**Writing – review & editing:** Jéssica de Oliveira Monteschio, Isabelle Naemi Kaneko, Laura Adriane de Moraes Pinto, Ana Guerrero, Gisele Maria Fagundes, James Pierre Muir.

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
