## [Decision Letter · Decision Letter 0]

15 Jan 2021

PONE-D-20-37779

Copaiba essential oil (Copaifera spp.) as natural preservative on sheep burgers: shelf life and consumer acceptability

PLOS ONE

Dear Dr. Monteschio,

Thank you for submitting your manuscript to PLOS ONE. After careful consideration, we feel that it has merit but does not fully meet PLOS ONE’s publication criteria as it currently stands. Therefore, we invite you to submit a revised version of the manuscript that addresses the points raised during the review process.

We look forward to receiving your revised manuscript.

Kind regards,

Christophe Hano

Academic Editor

PLOS ONE

2. Thank you for including your ethics statement:  "The project was approved by means of the Certificate of Presentation for Ethical Appreciation - CAAE (Certificate of Presentation of Ethical Appreciation) number 23282819.9.0000.5302 approved by the Research Ethics Committee Involving Human Beings.".   

Please provide additional details regarding participant consent. In the ethics statement in the Methods and online submission information, please ensure that you have specified (1) whether consent was informed and (2) what type you obtained (for instance, written or verbal, and if verbal, how it was documented and witnessed). If your study included minors, state whether you obtained consent from parents or guardians. If the need for consent was waived by the ethics committee, please include this information.

Reviewers' comments:

Reviewer's Responses to Questions

**Comments to the Author**

1. Is the manuscript technically sound, and do the data support the conclusions?

Reviewer #1: Partly

Reviewer #2: Yes

2. Has the statistical analysis been performed appropriately and rigorously? 

Reviewer #1: Yes

Reviewer #2: No

3. Have the authors made all data underlying the findings in their manuscript fully available?

Reviewer #1: Yes

Reviewer #2: Yes

4. Is the manuscript presented in an intelligible fashion and written in standard English?

Reviewer #1: Yes

Reviewer #2: Yes

5. Review Comments to the Author

Reviewer #1: MS : PONE-D-20-37779

COMMENTS:

1. Title needs to be changed into a continuous title, remove acceptability as it is understood that organoleptic properties are assessed when studying a food preservative

2. Abstract....... . Assessing during it shelf life the antioxidant activity, lipid oxidation, pH, color, cooking loss, texture and consumer acceptability. Rewrite this line ...

3. Mention species of Copaiba used for extraction of essential oil........this is very important as essential oil component changes with species...it should also reflect in title for the sake of future readers.

4. Material and methods

Give details of collection of plants or procurement of oil. If oil was extracted the give procedure and storage details...get plant species identified

Give at least minor procedural details of GC_MS and library used

Mention table 1 in GCMS section

How oil was used for treatment is not clear .please write it clearly

In result section the tables and figures should be arranged in sequential pattern. At several places there is incoherence leading to confusion.....

5. Discussion should be more critical citing other similar works on same commodity. As far as antioxidant property of oil is concerned almost all EO have it and this discovery is not new unless it is correlated to a specific species which is lacking.

6. Evolution of.....why Evolution ...i cannot understand...use some other term

7. Conclusion is just repetition of abstract....write a critical conclusion

8. Recheck the references and numbering

Reviewer #2: 1. The authors should describes more details about the experimental design particularly the concentrations tested such as BHT and CEO. In CEO treatments, why use 0.1 and 0.05 but not other concentrations?

2. Proper statistical analysis should be applied to the concentrations in Table 1.

3. Introduction section: Adds more contents about the practical use and literature of essential oils in the food industry. Adds a paragraph describing the constituents and efficacy of Copaiba essential oil.

4. Abstract: as possible replacement - as a possible replacement; Assessing during it shelf life - Assessing during its shelf life; essential oil showed antioxidant effect - essential oil showed an antioxidant effect; the fresh‐ like quality - the fresh‐like quality

5. Introduction: evaluating it influence - evaluating its influence

6. A representative chromatography spectra for chemical constituents of Copaiba essential oil is needed.

7. The authors should check grammar errors throughout the manuscript and improve them.

6. PLOS authors have the option to publish the peer review history of their article (what does this mean?). If published, this will include your full peer review and any attached files.

Reviewer #1: No

Reviewer #2: No

---

## [Author Response · Author response to Decision Letter 0]

10 Feb 2021

PONE-D-20-37779

Copaiba essential oil (Copaifera spp.) as natural preservative on sheep burgers: shelf life and consumer acceptability

Journal Requirements:

1. Please ensure that your manuscript meets PLOS ONE's style requirements, including those for file naming. The PLOS ONE style templates can be found at.

R: The manuscript has been revised and meets the PLOS ONE style requirements

2. Thank you for including your ethics statement: "The project was approved by means of the Certificate of Presentation for Ethical Appreciation - CAAE (Certificate of Presentation of Ethical Appreciation) number 23282819.9.0000.5302 approved by the Research Ethics Committee Involving Human Beings.". 

Please provide additional details regarding participant consent. In the ethics statement in the Methods and online submission information, please ensure that you have specified (1) whether consent was informed and (2) what type you obtained (for instance, written or verbal, and if verbal, how it was documented and witnessed). If your study included minors, state whether you obtained consent from parents or guardians. If the need for consent was waived by the ethics committee, please include this information.

R: The paper met all the requirements of the human ethics committee. Attached is a copy of the document presented to the research participants to request their consent, as well as the certificate of approval by the Ethics Committee.

R: The manuscript has been revised and meets the PLOS ONE style requirements

Specific comments:

 -Reviewer 1

1. Title needs to be changed into a continuous title, remove acceptability as it is understood that organoleptic properties are assessed when studying a food preservative

R: The title was changed. We hope you have answered the suggestions.

2. Abstract....... . Assessing during it shelf life the antioxidant activity, lipid oxidation, pH, color, cooking loss, texture and consumer acceptability. Rewrite this line ...

R: The sentence has been rewritten in more detail.

3. Mention species of Copaiba used for extraction of essential oil........this is very important as essential oil component changes with species...it should also reflect in title for the sake of future readers.

R: This information was added in the section “Preparation of burgers” and added to the title, in addition to the description of the essential oil extraction method. This information was important for the paper. Thank you for the consideration.

4. Material and methods

- Give details of collection of plants or procurement of oil. If oil was extracted the give procedure and storage details... get plant species identified

R: The information was added in the text, in the section “Preparation of burgers” and a better explanation was made.

- Give at least minor procedural details of GC_MS and library used

R: The required information has been added and more details have been added for GC_MS and more detailed methodology has been added for GC-FID.

- Mention table 1 in GCMS section 

R: As suggested by the reviewers, we inserted the representative chromatography spectra in the Material and Methods section referring to the GCMS in order to exemplify the chemical composition of the copaiba essential oil.

- How oil was used for treatment is not clear. Please write it clearly

R: The information was added in the text, in the section “Preparation of burgers” and a better explanation was made.

- In result section the tables and figures should be arranged in sequential pattern.

At several places there is incoherence leading to confusion.....

R: The order of the tables was verified and the error corrected and the order of the figures was adjusted, due to the inclusion of the chromatography spectrum.

5. Discussion should be more critical citing other similar works on same commodity. As far as antioxidant property of oil is concerned almost all EO have it and this discovery is not new unless it is correlated to a specific species which is lacking.

R: There are no studies in the literature regarding the inclusion of Copaiba essential oil in food and its effect on antioxidant activity, there are many studies in the health field, as a natural alternative in the prevention of diseases (proven with papers published in reputable journals, described so the research is of an innovative nature, showing an alternative in food that benefits human health.

“Effects of nanoemulsions prepared with essential oils of copaiba- and andiroba against Leishmania infantum and Leishmania amazonensis infections” - Experimental Parasitology. 2018.

“Copaiba oil-loaded commercial wound dressings using supercritical CO2: A potential alternative topical antileishmanial treatment” - The Journal of Supercritical Fluids. 2017.

“A synergistic nanoformulation of babassu and copaiba oils as natural alternative for prevention of benign prostatic hyperplasia” - Journal of Drug Delivery Science and Technology. 2018.

“Effects of a massage-like essential oil application procedure using Copaiba and Deep Blue oils in individuals with hand arthritis” - Complementary Therapies in Clinical Practice. 2018.

“Fast-Acting and Receptor-Mediated Regulation of Neuronal Signaling Pathways by Copaiba Essential Oil” - International Journal of Molecular Sciences. 2020.

6. Evolution of.....why Evolution ...i cannot understand...use some other term

R:The term "evolution" has been replaced by "changes" throughout the text.

7. Conclusion is just repetition of abstract....write a critical conclusion

R: The conclusion has been rewritten and changed.

8. Recheck the references and numbering

R: All references and numbering cited in the article were checked and updated, since other works were added in the introduction of the article, changing the numbers and the order of references in the work.

-Reviewer 2

1. The authors should describes more details about the experimental design particularly the concentrations tested such as BHT and CEO. In CEO treatments, why use 0.1 and 0.05 but not other concentrations?

R: The entire methodology was described in more detail in the section Material and methods - “Preparation of burgers”.

2. Proper statistical analysis should be applied to the concentrations in Table 1.

R: The quantitative analysis of essential oils is not strictly performed, due to the difficulty of obtaining analytical standards for all substances present in the essential oil. Thus, semi-quantitative analysis by area normalization was agreed. Thus, it is not possible to obtain data such as standard deviation, standard error of the media and others.

In several articles that characterize essential oils, the results are presented without statistical analysis values:

“Combined antioxidant and sensory effects of corn starch films with nanoemulsion of Zataria multiflora essential oil fortified with cinnamaldehyde on fresh ground beef pattise”- Meat Science. 2019.

Phytochemical Constituents and Antioxidant Activity of Sweet Basil (Ocimum basilicum L.) Essential Oil on Ground Beef from Boran and Nguni Cattle - International Journal of Food Science. 2019.

Antioxidant and Antibacterial Activity of Basil (Ocimum basilicum L.) Essential Oil in Beef Burger - Journal of Agricultural Science and Technology. 2015.

3. Introduction section: Adds more contents about the practical use and literature of essential oils in the food industry. Adds a paragraph describing the constituents and efficacy of Copaiba essential oil.

R: More information and new literature were added to the text.

4. Abstract: as possible replacement - as a possible replacement; Assessing during it shelf life - Assessing during its shelf life; essential oil showed antioxidant effect - essential oil showed an antioxidant effect; the fresh‐ like quality - the fresh‐like quality 

R: The errors have been corrected.

5. Introduction: evaluating it influence - evaluating its influence

R: The errors have been corrected.

6. A representative chromatography spectra for chemical constituents of Copaiba essential oil is needed.

R: A respresentative chromatograthy spectra for chemical constituents of Copaiba essential oil was added to the article.

7. The authors should check grammar errors throughout the manuscript and improve them.

R: Thank you for the consideration. Spelling errors were identified and the paper was sent to proofreading service. All necessary adjustment were made to improve the paper.

The paper has been professionally proofreading. The editorial certificate is below.

---

## [Decision Letter · Decision Letter 1]

1 Mar 2021

Effect of copaíba essential oil (Copaifera officinalis L.) as a natural preservative on the oxidation and shelf life of sheep burgers

PONE-D-20-37779R1

Dear Dr. Monteschio,

We’re pleased to inform you that your manuscript has been judged scientifically suitable for publication and will be formally accepted for publication once it meets all outstanding technical requirements.

Kind regards,

Christophe Hano

Academic Editor

PLOS ONE

Additional Editor Comments (optional):

Reviewers' comments:

Reviewer's Responses to Questions

**Comments to the Author**

1. If the authors have adequately addressed your comments raised in a previous round of review and you feel that this manuscript is now acceptable for publication, you may indicate that here to bypass the “Comments to the Author” section, enter your conflict of interest statement in the “Confidential to Editor” section, and submit your "Accept" recommendation.

Reviewer #1: All comments have been addressed

Reviewer #2: All comments have been addressed

2. Is the manuscript technically sound, and do the data support the conclusions?

Reviewer #1: Yes

Reviewer #2: Yes

3. Has the statistical analysis been performed appropriately and rigorously? 

Reviewer #1: Yes

Reviewer #2: Yes

4. Have the authors made all data underlying the findings in their manuscript fully available?

Reviewer #1: Yes

Reviewer #2: Yes

5. Is the manuscript presented in an intelligible fashion and written in standard English?

Reviewer #1: Yes

Reviewer #2: Yes

6. Review Comments to the Author

Reviewer #1: Accepted with some minor formatting according to Plos One policy. The authors be advised to recheck the language at proof stage. The MS is a nice piece of work with practical acceptability

Reviewer #2: (No Response)

7. PLOS authors have the option to publish the peer review history of their article (what does this mean?). If published, this will include your full peer review and any attached files.

Reviewer #1: No

Reviewer #2: No

---

## [Editor Report · Acceptance letter]

4 Mar 2021

PONE-D-20-37779R1 

Effect of copaíba essential oil (*Copaifera officinalis* L.) as a natural preservative on the oxidation and shelf life of sheep burgers 

Dear Dr. Monteschio:

I'm pleased to inform you that your manuscript has been deemed suitable for publication in PLOS ONE. Congratulations! Your manuscript is now with our production department. 

Kind regards, 

on behalf of

Dr. Christophe Hano 

Academic Editor

PLOS ONE